# l-Arginine and Beetroot Extract Supplementation in the Prevention of Sarcopenia

**DOI:** 10.3390/ph15030290

**Published:** 2022-02-26

**Authors:** Alfredo Córdova-Martínez, Alberto Caballero-García, Hugo J. Bello, Antoni Pons-Biescas, David C. Noriega, Enrique Roche

**Affiliations:** 1Biochemistry, Molecular Biology and Physiology, Faculty of Health Sciences, GIR Physical Exercise and Aging, University of Valladolid, Campus Duques de Soria, 42004 Soria, Spain; 2Department of Anatomy and Radiology, Faculty of Health Sciences, GIR Physical Exercise and Aging, University of Valladolid, Campus Los Pajaritos, 42004 Soria, Spain; alberto.caballero@uva.es; 3Department of Mathematics, School of Forestry, Agronomy and Bioenergy Engineering, GIR Physical Exercise and Aging, University of Valladolid, Campus Los Pajaritos, 42004 Soria, Spain; hjbello.wk@gmail.com; 4Research Group on Community Nutrition and Oxidative Stress, University of Balearic Islands, 07122 Palma de Mallorca, Balearic Islands, Spain; antonipons@uib.es; 5Department of Surgery, Ophthalmology, Otorhinolaryngology and Physiotherapy, Faculty of Medicine, Hospital Clínico Universitario de Valladolid, 03010 Valladolid, Spain; dcnoriega1970@gmail.com; 6Department of Applied Biology-Nutrition, Institute of Bioengineering, University Miguel Hernández, 03202 Elche, Spain; 7Alicante Institute for Health and Biomedical Research (ISABIAL Foundation), 03010 Alicante, Spain; 8CIBER Fisiopatología de la Obesidad y Nutrición (CIBEROBN), Instituto de Salud Carlos III (ISCIII), 28029 Madrid, Spain

**Keywords:** aging, creatine kinase, endurance, l-arginine, nitrate, speed, strength

## Abstract

Aging is associated with a significant decline in neuromuscular function, leading to a reduction in muscle mass and strength. The aim of the present report was to evaluate the effect of supplementation with nitric oxide precursors (l-arginine and beetroot extract) in muscular function during a training period of 6 weeks in elderly men and women. The study (double-blind, placebo-controlled) involved 66 subjects randomly divided into three groups: placebo, arginine-supplemented and beetroot extract-supplemented. At the end of this period, no changes in anthropometric parameters were observed. Regarding other circulating parameters, urea levels were significantly (*p* < 0.05) lower in women of the beetroot-supplemented group (31.6 ± 5.9 mg/dL) compared to placebo (41.3 ± 8.5 mg/dL) after 6 weeks of training. In addition, the circulating creatine kinase activity, as an index of muscle functionality, was significantly (*p* < 0.05) higher in women of the arginine- (214.1 ± 162.2 mIU/L) compared to the beetroot-supplemented group (84.4 ± 36.8 mIU/L) at the end of intervention. No significant effects were noticed with l-arginine or beetroot extract supplementation regarding strength, endurance and SPPB index. Only beetroot extract supplementation improved physical fitness significantly (*p* < 0.05) in the sprint exercise in men after 6 weeks (2.33 ± 0.59 s) compared to the baseline (2.72 ± 0.41 s). In conclusion, beetroot seems to be more efficient during short-term training while supplementing, preserving muscle functionality in women (decreased levels of circulating creatine kinase) and with modest effects in men.

## 1. Introduction

Aging, as a natural physiological process, is associated with advanced loss of neuromuscular function, leading to progressive disability and loss of independence, in parallel with a loss of skeletal muscle mass. This deterioration process known as sarcopenia is accompanied by alterations in the innervation of the central and peripheral nervous systems, hormonal changes, inflammatory processes and modification of food intake and nutrient assimilation [1,2,3,4,5]. The decrease in physical activity with age leads to sarcopenia, manifested by a progressive loss of muscle mass and strength [6]. This is accompanied by a high risk of adverse outcomes, more dependency, impaired quality of life, falls and fractures, hospitalization and even death [3]. During this deterioration process, muscle atrophy and myocyte death increase, explaining the loss in muscle mass [7].

At a molecular level, loss of muscle mass results from alterations in both synthesis and degradation of muscular proteins caused by multiple factors such as substrate availability, onset of inflammatory and oxidative stress, and satellite cell senescence, among others [8,9,10,11,12,13,14]. In this context, augmented production of nitric oxide (NO) can be positive at the muscle level by increasing blood flow that favors glucose uptake, contractility and activation of satellite cells to repair deteriorated fibers [9,15]. NO synthesis is carried out from l-arginine with the production of citrulline, by the activity of nitric oxide synthase (NOS). The NO-producing enzyme in muscle fibers is the neuronal isoform (nNOS). NO production affects muscle function, acting as a vasodilator [10].

Vasodilatory action of NO has been confirmed in animal models and in humans, favoring post-exercise recovery [16,17,18]. In this context, NO favors the elimination of waste metabolites (lactate and ammonia) resulting from muscular catabolism [19] that are related to fatigue appearance during exercise. Plasma nitrite levels increase following l-arginine supplementation in aerobic training routines [20,21]. However, plasma levels of l-arginine depend on the age of the individual, presenting elderly people with a more active catabolism [22]. Indirectly, NO could exert an antioxidant effect during exercise, allowing for an increased availability of circulating antioxidants due as well to its vasodilatory action [23].

Similar effects can be observed with the administration of molecules that are precursors or that stimulate NO production [24,25,26]. In this line, L-citrulline supplementation to active elders improves endothelial vascular function and muscle protein synthesis, complementing the effect of training [27]. In addition, the intake of sodium nitrate or nitrate-rich beetroot extract in athletes increases muscular energy efficiency, reducing pulmonary O_2_ uptake during submaximal exercise and enhancing athletic performance [21,28,29,30,31,32,33]. Moreover, the intake of sodium nitrate by patients affected by metabolic syndrome before low-intensity exercise attenuates the inflammatory and oxidative response to exercise and activates the expression of antioxidant genes and mitochondrial function [31,32]. Therefore, dietary nitrate, as a precursor of nitric oxide, is a promising molecule in which available evidence suggests potent physiological responses and benefits for clinical disorders such as hypertension, dementia, and sarcopenia [33].

Altogether, dietary supplementation with inducers of NO production, such as l-arginine or nitrates, shows positive results on muscle regeneration and repair, improving physical performance [16,34,35,36]. However, there are few studies devoted to this type of intervention in elders who are at clear risk of sarcopenia. The diagnosis of sarcopenia is mainly based on measures of muscle mass and strength, as well as other parameters defining physical fitness [3,37]. Total circulating proteins, urea, uric acid and creatinine levels, as well as the circulating urea/creatinine ratio are proposed as well as main markers of sarcopenia [38,39]. Therefore, the aim of the present report was to evaluate the effects of dietary supplementation with NO precursors such as l-arginine or nitrate (in the form of beetroot extract), during a training period (6 weeks), on the degree of muscle wasting in elderly men and women (over 60 years). Muscle mass and function were evaluated through the analysis of specific circulating parameters together with physical fitness probes compared to a placebo group.

## 2. Results

Anthropometric characteristics of the study participants are presented in Table 1.

The participants in the study were elderly men and women of similar age, with blood pressures within normal range (less than 140/90 mmHg in all cases) and with an almost healthy weight, according to the BMI. Men presented significantly higher values than women for weight, height and BMI at the beginning (Table 1). Meanwhile, percentage of body fat in men was significantly lower than in women (Table 1).

Supplementation with l-arginine or beetroot extract did not modify the anthropometric characteristics of the men and women during the six weeks of training (Table 2). No changes were observed for cardiovascular parameters (Table 2). Only heart rate was significantly lower in women supplemented with l-arginine compared to placebo (Table 2).

Table 3 shows the levels of circulating biochemical parameters as well as markers for muscle metabolism and mass for all participants, men and women. Serum levels of glucose, TP and markers of protein and purine catabolism, as well as CK, are within a healthy range. The levels of urea and creatinine are significantly higher in men than in women, although the urea/creatinine ratio and CK activity, as indexes of the level of muscle mass and functionality, are similar in both genders despite the fact that men have a higher percentage of lean mass than women.

Comparing supplementation, no significant differences in the initial values of circulating TP were noticed in the groups of women that ingested placebo, l-arginine or beetroot extract. However, after 6 weeks of dietary supplementation, the beetroot extract group had significantly lower serum levels of circulating TP than the placebo and l-arginine groups (Table 4). The women supplemented with beetroot extract presented significantly lower circulating levels of urea and uric acid than the placebo group, both at the beginning and at the end of intervention. In women, the urea/creatinine ratio of the group supplemented with beetroot extract was significantly lower than the urea/creatinine ratio of the group supplemented with l-arginine at the beginning of intervention and the placebo at the end of intervention (Table 4). Serum CK levels presented similar values at the beginning of intervention in the three groups (placebo, arginine and beetroot). However, at the end of the intervention period, CK significantly doubled the value in the group of women supplemented with beetroot. Conversely, circulating TP and creatinine levels in men displayed significant differences in the beetroot group at the beginning of intervention; thus, they cannot be due to supplementation. The remaining circulating parameters in the supplemented groups did not show significant differences compared to the placebo group (Table 4).

The main finding is that supplementation of women with beetroot extract tends to decrease circulating TP levels and increase creatinine levels after 6 weeks of training. In addition, supplementation with l-arginine tends to increase serum CK levels compared to placebo and beetroot extract-supplemented group, although the basal values of the l-arginine-supplemented group were significantly higher (Table 4).

Table 5 shows the results of the physical activity tests at the beginning of the training period. Men and women participating in the study maintain an active lifestyle and perform regular physical activity (around 10 h/week) in gyms. This observation explains the robust initial values of physical fitness markers regarding strength, endurance and speed (included in the SPPB). No significant differences were observed in the balance test (included in the SPPB), with a maximal score for all participants (not shown). Collectively, the positive scores were reflected in an SPPB index > 10, above the cut-off value for the diagnosis of sarcopenia. Men presented significantly higher strength and endurance than women, although there were no significant differences in the speed test, squat test or SPPB index, indicating similar levels of low sarcopenia in the men and women who participated in the study.

Regarding supplementation, no significant differences were observed in women or men for strength, endurance and SPPB index after 6 weeks of training (Table 6). However, the time to run 4 m in men was reduced from baseline in all groups at the end of intervention. Nevertheless, the speed time was significantly longer in the beetroot-supplemented group compared to placebo or l-arginine-supplemented groups. A similar effect on running speed was also observed in women consuming beetroot extract, but before taking the supplement, while no significant differences were observed at the end of intervention. Altogether, the most significant change was the improvement in running speed in men after supplementation with beetroot extract compared to the beginning, while no significant effects were observed in strength and endurance. Nevertheless, beetroot-supplemented men started at a much slower pace than the placebo and arginine-supplemented groups. Therefore, one possible explanation is that they have more margin to improve in this test.

## 3. Discussion

Sarcopenia is a geriatric syndrome defined as a progressive and generalized loss of skeletal muscle mass and strength [1,2,3,4,5]. Regarding the physical activity tests used for the assessment of sarcopenia and taking into account the cut-off points for grip strength and the SPPB index, the men and women participating in this trial have no obvious symptoms of sarcopenia [40]. Participants were elders (more than 60 years) at risk of reduced muscle mass as a result of muscle aging.

Resistance training is an effective treatment to optimize body fat mass, improve muscle strength and performance, and maintain muscle mass and flexibility in healthy elders, delaying sarcopenia [41]. In our study, we did not observe any effect on the fitness parameters in women or men who participated in the proposed training program. This aspect is difficult to manage in elderly populations compared to younger athletes. Physiological and psychological response at an old age range vs. young people is not comparable. Training programs, referring to the level of physical fitness achieved, are of lesser impact, compared to designs of greater intensity performed in young athletes. The physical effort made by elders during the training program was lower in intensity than programs of young athletes, designed to undergo adaptive remodeling and new functional and metabolic demands [42]. In this line, no significant changes after the training program were observed, except for gait speed in men supplemented with beetroot extract at the end of intervention compared to the beginning. A possible explanation is that dietary supplementation with nitrate, administered in the form of beetroot extract, improves skeletal muscle contractile function and thereby improves muscle power and sprint performance [43]. A possible mechanism by which nitrate acts to improve muscle metabolic efficiency response involves increased NO levels that may attenuate the ATP/phosphocreatine cost associated with skeletal muscle force production [44]. Nevertheless, an alternative mechanism proposes that nitrate directly alters mitochondrial efficiency through attenuation of the ATP/oxygen ratio [45]. Another possible explanation is that 6 weeks of training may be too short to obtain muscle benefits, with this point as a limitation of the study.

Conversely, the improvement in muscle function in the beetroot extract-supplemented group of women is associated with an increase in serum creatinine levels and a decrease in circulating TP. High plasmatic TP levels as well as low creatinine concentration have been shown to be associated with sarcopenia and loss of muscle mass and function in both men and women [46]. In addition, serum plasma albumin level is a biomarker for predicting the effects of low-load resistance training programs on muscle hypertrophy in elders [47]. Therefore, the effects of beetroot extract supplementation vs. placebo in women during 6 weeks of training on serum TP and creatinine levels could be indicative of a tendency to increase muscle mass [48]. In this context, it has been demonstrated that endogenous NO is closely involved in the induction of skeletal muscle fiber hypertrophy by reducing protein degradation and increasing protein synthesis [49,50]. In addition, circulating urea and uric acid can be considered as markers of protein and purine catabolism, respectively. Altogether, these observations suggest that women in the beetroot extract-supplemented group have lower rates of protein and purine catabolism already at the beginning of intervention compared to placebo or l-arginine groups. A significant interaction between supplementation and training is noteworthy in the serum TP and creatinine levels of women of the beetroot group, increasing significantly after 6 weeks of intervention. This interaction (supplement × training) was not noticed in women supplemented with l-arginine or placebo (Table 4).

Supplementation with l-arginine during 6 weeks of training maintained the baseline fitness parameters of women and men at virtually the same level as the placebo group. However, a serum marker of muscle function, such as CK activity, increases with l-arginine supplementation after 6 weeks of training, particularly in women, indicating a possible improvement in muscle function and aging status. CK is an enzyme present in heart and skeletal muscle. Since participants did not present cardiac pathologies (see exclusion criteria), CK serves as an indicator of the level of muscle damage. It is possible that the CK increase observed in the arginine group might be due to an increase in work intensity as a consequence of improvement in the perception of physical pain and in the decrease in sensation of tiredness mediated by the supplement. This hypothesis needs confirmation in further experiments. The release of NO mediated by arginine produces an improvement in tissue perfusion through mechanisms of vasodilatation and angiogenesis, and these changes lead to a better supply of nutrients (glucose, fatty acids and amino acids) [50,51]. In this context, cellular ATP production is improved, likely activating mTOR-dependent anabolic pathways that promote protein synthesis. This increased nutrient supply also enhances lipolysis and glucose management, thus preventing muscle fat infiltration [51,52]. In addition, CK levels are significantly decreased in sarcopenia with respect to non-sarcopenic individuals [39]. Therefore, the increase in this muscle enzyme in the group of women who ingested l-arginine during the 6-week training period could be compatible with a more functional muscle mass [39].

Regarding the two endogenous NO-generating pathways, nNOS (l-arginine supplementation) and direct NO production by nitrate–nitrite intake (beetroot supplementation), the one that has shown the greatest significant effects on sprint exercise and muscle mass has been the beetroot extract. However, dietary supplementation with l-arginine improves muscle function by increasing circulating levels of CK, likely as a consequence of NO production. A large body of evidence now suggests that regulation of nNOS is an important part of mammalian skeletal muscle adaptation to intense endurance exercise. Most human studies suggest that endurance exercise leads to an increase in nNOS expression [53,54,55]. Therefore, increased basal expression of nNOS appears to be an adaptation to endurance training in humans. The availability of l-arginine may also influence the rate of NO production in elderly women and men, as evidenced by the increased muscle function of participants supplemented with l-arginine during the training period.

Limitations to consider for future research include: longer interventions than 6 weeks in order to check possible improvements in muscle function, adaptation of each exercise routine to the particular characteristics of each individual (the exercise program designed in the present report was general for the whole group), effects of the supplements + exercise routine in advanced cases of sarcopenia and other pathologies frequent in elders. 

## 4. Materials and Methods

### 4.1. Study Design and Participants

The trial was a double-blind, placebo-controlled study. Sixty-six subjects (22 women and 44 men) randomly divided into three groups (placebo, arginine-supplemented and beetroot-supplemented) participated in the study. Supplements were provided in the form of capsules. All groups followed the same diet (55% carbohydrates, 30% lipids and 15% proteins), adjusting caloric intake to the daily activity and exercise expenditure. Gender differences were considered in each group.

The study was approved by the Ethics Committee of Hospital General de Alicante (Spain) (Ref. CEIm PI2019-070). Prior to starting the intervention, participants signed the written informed consent. The inclusion criteria were to be over 60 years of age at the end of intervention and to display a healthy status, not affected by the following exclusion criteria: history of dementia, severe or moderate chronic obstructive pulmonary disease (COPD) with Bodex (COPD assessment index) C or D, functional limitation according to Barthel scale less than 100 (maximum score) and autonomy according to Lawton–Brody scale less than 8 (maximum value), recent acute myocardial infarction (3 to 6 months), unstable angina, uncontrolled atrial or ventricular arrhythmias, dissecting aortic aneurysm, severe aortic stenosis, acute endocarditis/pericarditis, uncontrolled arterial hypertension (>100–180 mmHg), acute thromboembolic disease, acute/chronic heart failure with NYHA > II (scale of the New York Heart Association), acute/chronic respiratory failure, uncontrolled orthostatic hypotension, diabetes mellitus with acute decompensation or uncontrolled hypoglycemia, and recent fracture (last month). Additional exclusion criteria were previous supplementation with amino acids or other nutritional compounds to improve physical performance or any other circumstances that the physician may consider as limiting physical activity. 

The week prior to intervention, all the proposed physical and clinical laboratory tests were performed (see below). These tests were repeated at the end of the 6 weeks of intervention. The activities performed in each of the training sessions were recorded in order to analyze the type of activity and to assess the level of intensity. All training sessions were structured in 3 blocks: warm-up, main part (strength and endurance work) and cool-down. In this way, we adjusted the work times in each of the groups and made a homogeneous program for all participants.

### 4.2. Supplementation

Initially, at the beginning of each week, the doctor responsible for the research team distributed the corresponding l-arginine, beetroot extract and placebo capsules in numbered bags hermetically sealed. Capsules were taken during the 6-week follow-up period. Content of each capsule was: 3 g of l-arginine and 3–3.5 g of dry beetroot extract, standardized to contain approximately 0.3% of betaine as a precursor of 100 mg of nitrate, according to [56] and [57], respectively. Supplements were purchased from Lanier-Pharma SL (Madrid, Spain).

All capsules displayed a similar appearance. The different compounds were identified with a code on the package containing a number plus the letters A, B or C. During the intervention period, none of the study participants nor the scientific research team knew the composition of the capsules. The l-arginine and beetroot extract capsules (1 daily) contained lactose and starch in addition to the corresponding active ingredient. Placebo capsules had the same weight and appearance of those with the active ingredients but contain only lactose and starch. 

### 4.3. Anthropometry

The first and last day of the intervention and 15 min after blood extraction (8 a.m.), anthropometric parameters were determined. Height and weight were recorded to obtain body mass index (BMI), and percentage of body fat (%) was assessed with the Tanita BC-418MA bioimpedance meter (Madrid, Spain). Blood pressure was measured with the digital sphygmomanometer OMRON M2 Basic HEM-7120 (Madrid, Spain). 

### 4.4. Physical Activity Tests

For the assessment of physical capacities, validated different tests were carried out. The following aspects were assessed: A) hand grip strength using a JAMAR digital dynamometer from 0–90 kg (Performance Health, Warrenville, IL, USA); B) 6 min test (endurance), which was carried out on a 400 m athletics track approved by the Spanish athletics federation (RFEA), measuring the distance covered in these 6 min; C) Short Physical Performance Battery (SPPB) frailty test, consisting of 3 tests measuring gait speed, balance in three different positions and leg strength (squat test). In the walking gait speed test, the participant walks a distance of 4 m monitored by photoelectric cells at both ends. Since this test may involve variability, to minimize it, the volunteers were asked to start walking 5 m before timing at their usual walking speed. In the balance test, the participant attempts to maintain 3 positions: feet together, semi-tandem and tandem, at 10 s for each one. Finally, in the squat test, the participant stands up and sits down on a chair 5 times without the aid of the arms, as fast as possible, and the total time is recorded. All individuals used the same model of chair with a height of 45 cm to the floor. The sum of the scores obtained determines the frailty level of each subject. Each test is scored from 0 (worst performance) to 4 (best performance). The balance test is scored according to a hierarchical combination of performance in the 3 component subtests. For the other 2 tests, a score of 0 is assigned to those who do not complete or attempt the task and scores from 1 to 4 according to the time invested. Scores are as follows: 4 points for subjects who completed the entire exercise in less than 11.16 s, 3 points for a time between 11.20–13.69 s, 2 points for a time between 13.70–16.69 s, 1 point for a time more than 16.70 s. A sum of scores below 10 predicts the risk of frailty, disability and sarcopenia. The initial values of the physical tests do not show a clear establishment of sarcopenia. However, participants are part of a population segment with a clear risk of developing sarcopenia due to their age [58].

### 4.5. Training

The training sessions were monitored by one of the members of the research team, participating at the same time. At the end of each training session, the researcher proceeded to collect the information. The type and intensity of the exercises and the subjective feeling of each participant were determined. The distribution of times and workloads is summarized in Table 7.

### 4.6. Blood Analysis

The blood extractions were performed the week before the intervention and on the final day. The extractions were performed at 08:00 a.m. after 12 h of fasting and 8 h of resting in a certified laboratory from the Spanish Health System. Blood volume was 10 mL in “vacutainer” tubes for serum, and 5 and 3 mL in tubes with EDTA for plasma. Patients remained seated and quiet during the extraction process. Then, tubes were stored in refrigerated containers at 4 °C. 

Markers of protein catabolism, purines and muscle mass (creatine kinase/CK, creatinine, glucose, urea, uric acid and total proteins/TP) were determined with the Architect ci8200^®^ analyzer (Abbott, Madrid, Spain). 

### 4.7. Statistics

Statistical analyses were performed using IBM Statistical Package (SPSS Version 24, New York, NY, USA). Data were expressed as mean ± standard error of the mean (x ± SEM). After verifying normal distribution using the Shapiro–Wilk test, Levene’s homogeneity of variances was studied. A two-factor ANOVA test was used to compare the groups, the training factor (T) and the supplementation factor (S). One-way ANOVA tests were performed on those variable factors that showed a level of statistical significance in order to know which groups showed significant differences. For those parameters that did not show a normal distribution (i.e., SPPB), the Kruskal–Wallis test was performed. In all calculations, the level of statistical significance was *p* < 0.05.

## 5. Conclusions

In summary, supplementing the diet of elderly women and men during a training period with beetroot extract improves muscle function mainly in elderly women but does not improve general fitness status in elderly women and men, except for gait speed. l-arginine supplementation produces less significant effects.

## Figures and Tables

**Table 1 pharmaceuticals-15-00290-t001:** Anthropometric parameters and blood pressure values of the women and men at the beginning of the study.

Parameter	Women	Men	* *p*
*n*	22	44	
Age (years)	63.8 ± 4.6	65.9 ± 6.4	
Weight (kg)	65.8 ± 8.2	80.8 ± 13.0	* 0.000
Height (cm)	162 ± 7	172 ± 7	* 0.000
Body mass index (kg/m^2^)	25.2 ± 2.6	27.4 ± 3.6	* 0.032
Fat (%)	32.8 ± 5.5	25.0 ± 5.1	* 0.000
Lean mass (kg)	42.7 + 7.8	58.0 + 8.1	
Heart Rate (beats/min)	69.3± 10.0	70.3 ± 13.1	
Systolic blood pressure (mmHg)	127 ± 15	130 ± 21	
Diastolic blood pressure (mmHg)	78.8 ± 11.7	± 7.8	

* Statistical significance.

**Table 2 pharmaceuticals-15-00290-t002:** Effect of l-arginine or beetroot extract supplementation on anthropometric and cardiovascular parameters of men and women trained for six weeks.

Parameter	Moment	PL	ARG	BEET	ANOVA (*p*)
WOMEN (*n*)		8	7	7	
Weight (kg)	Initial	64.7 ± 9.9	62.3 ± 4.9	68.1 ± 7.7	
	Final	64.5 ± 9.9	64.9 ± 4.8	69.9 ± 10.1	
Fat (%)	Initial	33.2 ± 6.0	29.9 ± 5.9	33.6 ± 5.1	
	Final	31.7 ± 5.4	30.8 ± 5.5	33.3 ± 5.6	
Lean mass (kg)	Initial	41.8 ± 6.8	42.6 ± 6.2	43.4 ± 5.9	
	Final	42.1 ± 7.3	43.1 ± 6.5	42.9 ± 6.2	
Heart Rate	Initial	72.3 ± 14.1	66.8 ± 13.1	68.2 ± 3.5	
(beats/min)	Final	73.4 ± 12.8	60.0 ± 3.7*	67.4 ± 3.6	S (0.034)
Systolic BP	Initial	131 ± 11	121 ± 17	126 ± 17	
(mmHg)	Final	136 ± 8	124 ± 15	127 ± 15	
Diastolic BP	Initial	81.3 ± 12.7	71.8 ± 9.5	79.8 ± 11.5	
(mmHg)	Final	82.9 ± 10.5	74.2 ± 10.4	81.2 ± 9.3	
MEN (*n*)		14	15	15	
Weight (kg)	Initial	77.1 ± 8.9	81.7 ± 13.3	82.1 ± 14.1	
	Final	74.4 ± 5.9	72.2 ± 2.9	81.6 ± 14.5	
Fat %	Initial	20.5 ± 2.8	24.9 ± 5.4	25.9 ± 5.1	
	Final	20.9 ± 3.5	27.1 ± 8.1	24.1 ± 6.9	
Lean mass (kg)	Initial	58.4 ± 7.2	59.1 ± 6.9	58.0 ± 8.1	
	Final	56.2 ± 7.7	53.6 ± 7.1	58.5 ± 6.7	
Heart Rate	Initial	57.3 ± 12.1	71.7 ± 21.5	63.6 ± 9.5	
(beats/min)	Final	56.0 ± 12.5	66.7 ± 5.1	66.9 ± 9.2	
Systolic BP	Initial	133.6 ± 16.2	127 ± 14.4	141.6 ± 13.9	
(mmHg)	Final	135.6 ± 12.3	131.9 ± 14.8	132.1 ± 20.4	
Diastolic BP	Initial	81.0 ± 5.3	81.0 ± 4.4	84.4 ± 9.0	
(mmHg)	Final	87.3 ± 3.8	71.3 ± 11.9	82.0 ± 10.9	

Abbreviations used: ARG, arginine-supplemented group; BEET, beetroot extract-supplemented group; BP, blood pressure; PL, placebo. (*) indicates significant difference (*p* < 0.05) of supplementation (S) with respect to placebo after one-way ANOVA analysis.

**Table 3 pharmaceuticals-15-00290-t003:** Biochemical parameters related to muscle mass and function separated by gender (men and women).

Parameter	Women	Men	* *p*
Glucose (mg/dL)	89.2 ± 1.9	89.1 ± 2.6	
Total protein (g/dL)	7.2 ± 0.4	7.2 ± 0.3	
Urea (mg/dL)	35.5 ± 10.8	42.5 ± 9.1 *	* 0.005
Creatinine (mg/dL)	0.75 ± 0.12	0.84± 0.12 *	* 0.000
Urea/Creatinine	47.8 ± 2.7	51.5 ± 3.6	
Uric acid (mg/dL)	4.3 ± 0.6	5.2 ± 0.5	
Creatine kinase (mIU/L)	107.0 ± 53.2	100.1 ± 39.4	

* Statistical significance.

**Table 4 pharmaceuticals-15-00290-t004:** Levels of circulating parameters in the placebo, arginine- and beetroot-supplemented groups in women and men participating in the study.

Parameter	Moment	PL	ARG	BEET	ANOVA (*p*)
WOMEN					
Glucose	Initial	88.9 ± 12.1	87.6 ± 9.1	90.0 ± 9.6	
(mg/dL)	Final	89.0 ± 10.9	81.8 ± 12.7	90.1 ± 9.7	
Total protein	Initial	7.3 ± 0.4	7.2 ± 0.2	7.2 ± 0.4	S (0.012)
(g/dL)	Final	7.4 ± 0.4	7.1 ± 0.1	6.7 ± 0.4 *^#^	S x T (0.023)
Urea	Initial	39.4 ± 11.2	40.0 ± 13.9	30.6 ± 7.7 *	S (0.006)
(mg/dL)	Final	41.3 ± 8.5	36.8 ± 7.1	31.6 ± 5.9 *	
Creatinine	Initial	0.8 ± 0.2	0.7 ± 0.1	0.8 ± 0.1	S (0.040)
(mg/dL)	Final	0.7 ± 0.1	0.7 ± 0.1	0.9 ± 0.1 *^#^	S × T (0.035)
Urea/Creatinine	Initial	50.5 ± 10.3	58.9 ± 21.9	41.1 ± 11.4 ^#^	S (0.000)
	Final	58.8 ± 14.9	49.6 ± 6.3	36.9 ± 8.1 *	
Uric acid	Initial	5.0 ± 1.2	4.0 ± 0.2 *	3.8 ± 0.5 *	S (0.000)
(mg/dL)	Final	5.0 ± 0.8	4.2 ± 0.3	3.8 ± 0.8 *	
Creatine kinase	Initial	123.0 ± 58.1	123.0 ± 64.3	87.4 ± 43.0	S (0.009)
(mIU/L)	Final	103.2 ± 43.0	214.1 ± 162.2 *	84.4 ± 36.8 ^#^	
MEN					
Glucose	Initial	91.0 ± 5.2	84.0 ± 1.0	89.6 ± 6.6	
(mg/dL)	Final	81.3 ± 8.1	81.3 ± 4.7	89.8 ± 12.2	
Total Protein	Initial	6.8 ± 0.2	6.9 ± 0.1	7.3 ± 0.2 *	S (0.002)
(g/dL)	Final	7.1 ± 0.3	6.8 ± 0.2	7.1 ± 0.3	
Urea	Initial	45.7 ± 13.8	44.0 ± 11.1	41.0 ± 7.2	
(mg/dL)	Final	40.7 ± 3.5	44.0 ± 5.0	41.6 ± 7.6	
Creatinine	Initial	0.8 ± 0.2	1.0 ± 0.1	0.8 ± 0.1 ^#^	S (0.007)
(mg/dL)	Final	0.9 ± 0.1	1.1 ± 0.1	0.9 ± 0.1	
Urea/Creatinine	Initial	57.1 ± 25.2	44.0 ± 16.3	50.8 ± 10.0	
	Final	48.1 ± 7.5	41.3 ± 4.7	46.8 ± 13.4	
Uric acid	Initial	5.1 ± 0.5	5.2 ± 0.4	5.3 ± 0.7	
(mg/dL)	Final	5.7 ± 0.6	5.6 ± 0.8	5.6 ± 0.8	
Creatine Kinase	Initial	135.0 ± 33.2	80.0 ± 30.3	96.0 ± 36.9	
(mIU/L)	Final	133.1 ± 73.0	114.2 ± 59.1	130.3 ± 53.8	

Abbreviations used: ARG, arginine-supplemented group; BEET, beetroot extract-supplemented group; BP, blood pressure; PL, placebo. (*) Indicates significant differences compared to placebo. (^#^) Indicates significant differences comparing ARG vs. BEET. Significance (*p* < 0.05) determined after one-way ANOVA analysis. S indicates a significant effect of supplementation; T indicates a significant effect of training. S × T indicates an interaction between supplementation and training. Significance (*p* < 0.05) was determined after two-way ANOVA analysis.

**Table 5 pharmaceuticals-15-00290-t005:** Parameters of physical activity tests at the beginning of intervention.

Parameter	Women	Men	* *p*
Strength (Kg)	28.8 ± 8.0	44.5 ± 7.0 *	* 0.000
Endurance (m)	794 ± 132	967 ± 211 *	* 0.000
Speed (s)	2.46 ± 0.50	2.52 ± 0.44	
Squat (s)	11.4 ± 1.8	11.0 ± 2.0	
SPPB index (range)	11.3 ± 0.6 (11.1–11.5)	11.5 ± 0.7 (11.2–11.8)	

* Statistical significance.

**Table 6 pharmaceuticals-15-00290-t006:** Physical activity parameters in the studied groups.

Parameter	Moment	PL	ARG	BEET	ANOVA (*p*)
WOMEN					
Strength (Kg)	Initial	26.7 ± 4.2	27.8 ± 3.3	30.8 ± 10.9	
	Final	27.2 ± 2.7	28.8 ± 3.3	33.5 ± 9.6	
Endurance (m)	Initial	776 ± 131	883 ± 125	771 ± 132	
	Final	792 ± 142	912 ± 223	834 ± 142	
Speed (s)	Initial	2.27± 0.16	2.23 ± 0.10	2.70 ± 0.65 *	S (0.030)
	Final	2.37 ± 0.26	2.19± 0.47	2.52 ± 0.48	
Squat (s)	Initial	12.5 ± 1.5	11.4 ± 1.25	10.7 ± 1.9	
	Final	11.1 ± 1.8	11.6 ± 3.7	10.4 ± 1.3	
SPPB index	Initial	11.1 ± 0.6	11.4 ± 0.5	11.4 ± 0.7	
(points)	Final	11.3 ± 0.5	11.8 ± 0.4	11.7 ± 0.5	
MEN					
Strength (Kg)	Initial	44.0 ± 6.0	47.7 ± 2.51	43.0 ± 8.2	
	Final	48.3 ± 5.8	40.0 ± 12.1	44.4 ± 8.0	
Endurance (m)	Initial	1050 ± 229	1008 ± 343	914 ± 169	
	Final	1102 ± 226	949 ± 164	1049 ± 155	
Speed (s)	Initial	2.03± 0.02	2.18 ± 0.07	2.72 ± 0.41 *^#^	S (0.025)
	Final	1.95 ± 0.26	2.27± 0.37	2.33 ± 0.54 *^&^	
Squat (s)	Initial	8.9 ± 0.5	10.7 ± 1.8	11.6 ± 1.7	
	Final	9.4 ± 1.5	10.9 ± 1.2	10.1 ± 1.7	
SPPB index	Initial	12.0 ± 0.0	11.7 ± 0.6	11.3 ± 0.8	
(points)	Final	87.3 ± 3.8	71.3 ± 11.9	82.0 ± 10.9	

Abbreviations used: ARG, arginine-supplemented group; BEET, beetroot extract-supplemented group; BP, blood pressure; PL, placebo. PL: placebo. (*) Indicates significant differences compared to placebo. (#) Indicates significant differences comparing ARG vs. BEET. (&) Indicates significant differences comparing initial vs. final situations. Significance (*p* < 0.05) determined after one-way ANOVA analysis. S indicates a significant effect of supplementation.

**Table 7 pharmaceuticals-15-00290-t007:** Physical activity protocol performed in each training session.

Session Structure	Time (min)	Protocol	Effort Level *
Warm-up	10	General mobility, light movements	4
Balance	5	Standing and monopodal exercises	3
Aerobic endurance	10	Walking, slow running	7
Aerobic endurance	20	Overload exercises, with balls, dumbbells, rubber bands, steps	8

(*) Based on the Borg scale adapted to elders [59].

## Data Availability

Data are contained within the article.

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
