# Peer review of "l-Arginine and Beetroot Extract Supplementation in the Prevention of Sarcopenia"

_pharmaceuticals, 2022, doi:10.3390/ph15030290_

Round 1
Reviewer 1 Report
This study aimed to evaluate the effect of supplementation with nitric oxide precursors (L-arginine and beetroot extract) in muscular function during a training period of 6 weeks in elder men and women. This is a very great study, thus I have major revisions prior the publication.
-in the abstract: to add the statistical and numbers in the results.
-in the introduction: is too long and must be reduced. In addition, a reference and informmations regarding to methods to quantify the sarcopenia will improve the introduction. Moreover, the hypothesis of study is not clear and requires a fine highlights about that.
-In the methods: please: to add only elderly (over 60y).
-In the results section: please to add all p values.
The analysis must be conducted evaluating only elderly (over 60y old).
In the table 6, what are the p values of strenght and endurance tests? p= 0.000 is infinite... In addition, data of food intake are crucial to discuss these findings since the dietary food intake is associated with sarcopenia.
-In the discussion: Must be reformulated considering the new data with assessing of only elderly.
-
Author Response
REVIEWER 1:
This study aimed to evaluate the effect of supplementation with nitric oxide precursors (L-arginine and beetroot extract) in muscular function during a training period of 6 weeks in elder men and women. This is a very great study, thus I have major revisions prior the publication.
Answer: We appreciate these comments and we provide the answers to the points raised by the reviewer in the following section.
- In the abstract: to add the statistical and numbers in the results.
Answer: Significance (p<0.05) and values of the parameters mentioned have been included in the Abstract. See lanes 31-34 and 37-38.
- In the introduction: is too long and must be reduced. In addition, a reference and informations regarding to methods to quantify the sarcopenia will improve the introduction. Moreover, the hypothesis of study is not clear and requires a fine highlights about that.
Answer: The length of Introduction has been shortened. The hypothesis and methods to determine sarcopenia are highlighted at the end of this section (lanes 87-99).
- In the methods: please: to add only elderly (over 60y).
Answer: One of the main inclusion criteria is to be over 60 years of age at the end of the intervention (lanes 112-113).
- In the results section: please to add all p values.
Answer: The p values of significant parameters have been included in Tables 3, 5 and 7.
The analysis must be conducted evaluating only elderly (over 60y old).
Answer: The main part of participants were 60 years old at the beginning and at the end of intervention. Only 2 participants started with 59 years, but were at 60 after 2 and 3 days of intervention. Therefore, we can consider that the analysis is mainly conducted in elderly over 60 years. We believe that a period of 2-3 days to be included in the group of 60 years are not generating a bias in the statistics.
In the table 6, what are the p values of strenght and endurance tests? p= 0.000 is infinite... In addition, data of food intake are crucial to discuss these findings since the dietary food intake is associated with sarcopenia.
Answer: A p=0.000 means that the probability that the compared values are different is very high (99.999%). Unfortunately, the statistics program does not give more decimals. In this line, a p=0.032 (i.e. Table 2 for BMI values) indicates that the probability that these values are different is 96.8%. We cannot be more precise regarding the p values that we have maintained in Tables 2, 4 and 6. On the other hand, diet composition is indicated in Materials and Methods (lanes 106-107). We agree that diet plays a key role in sarcopenia prevention. Since all participants followed the same diet, this variable does not seem to be determinant in the final results obtained in this particular intervention.
- In the discussion: Must be reformulated considering the new data with assessing of only elderly.
Answer: Discussion has been adapted to the data generated in elders.
Reviewer 2 Report
Reviewer Recommendation and Comments for Manuscript Number:
pharmaceuticals-1590928
“L-Arginine and Beetroot Extract Supplementation in the Prevention of Sarcopenia”
These are my general/specific comments:
GENERAL:
The current manuscript is a very interesting and well-written original manuscript, yet there are limitations to the manuscript in its current form. The researchers evaluated the effects of dietary supplementation of nitric oxide precursors, L-arginine or beetroot extract as well as control, in aging humans that underwent a six-week training period. Overall, the manuscript has relatively minor editorial (grammar, language, etc.,) modifications that should be addressed. The authors should provide explicit hypothesis(es) tested as well as provide evidence for purity of supplements utilized. Even though these are among limitations in the manuscript’s present form, the authors should be able to address these points for full consideration of acceptance.
SPECIFIC:
TITLE
- None
ABSTRACT
- Page 1, Line 37: Change "...to the beginning." to "...to baseline."
- Page 1, Line 37: Add: ..."during short-term training while supplementing, preserving..." following "supplement"
INTRODUCTION
- Pages 1-2, Line s 45-52: "Degeneration" has a very specific denotative meaning in medicine/physiology; and, while the author's note the disability and loss of skeletal muscle mass that occurs with the aging process, it is not putative that the "hallmarks" of the aging process include "degeneration". The word "deterioration" would be a better word choice.
- Page 3, Line 108: The authors should state explicit/specific hypothesis(es)
MATERIALS and METHODS
- Page 3, Line 129: Please explain/clarify this range for arterial hypertension given.
- Page 3, Line 137: Include "to" between "prior intervention"
- Page 3, Line 143: Change "...for all of them." to "...for all partcipants."
- Page 3, Line 145: Change "From the beginning of the study and at the..." to "Initially, at the..."
- Page 4, Line 150: Were supplements/compounds USP grade? If not, how was the purity verified?
- Page 4, Line 175: Change "tries" to "attempts"
- Page 4, Line 192: Change "he" to "the researcher"
RESULTS
- Page 5, Lines 202-203: Change "...and the second, the last day." to "...the second day, and on the final day."
- Page 5, Lines 203-205: The authors should describe how the blood draw site was prepared (i.e. sterile prep, etc.,) prior to extraction.
- Pages 5-6, Lines 229-230: What does "an almost healthy weight" signify?
- Page 7, Line 259: Delete "have"
- Page 7, Line 272: Change "rest of" to "remaining"
- Page 7, Line 273: Change "taking the supplements" to "supplemented"
- Page 8, Line 293-294: Change "optimal" to "robust" or another term the authors feel is appropriate
- Page 8, Line 296: Change "All these positive scores..." to "Collectively, the positive scores..."
- Page 8, Line 305: Change "on" to "for"
DISCUSSION
- Page 9, Line 335: Change "...of the women and men who..." to "...in women or men who..."
- Page 9, Lines 340-341: How can the authors make this statement, for they did not measure intensity directly nor do they have a young group included in the current study. This must be restated.
- Page 10, Lines 345-349: Six weeks of exercise training may be too short of an exposure duration (possible limitation?).
- Page 10, Line 368: Delete "all"
- Page 10, Line 368: Change "suggests" to "suggest"
- Page 10, Line 382: Change "...level of muscle use and damage" to "...level of muscle damage.". It does not indicate the "level of use", which would be indicative of activation.
- Page 11, Paragraph 1, Line 408: The authors should note/address the/any limitation(s) of the current study.
REFERECES
- None
FIGURE LEGENDS/FIGURES
- None
Author Response
REVIEWER 2:
Reviewer Recommendation and Comments for Manuscript Number: pharmaceuticals-1590928
“L-Arginine and Beetroot Extract Supplementation in the Prevention of Sarcopenia”
These are my general/specific comments:
GENERAL:
The current manuscript is a very interesting and well-written original manuscript, yet there are limitations to the manuscript in its current form. The researchers evaluated the effects of dietary supplementation of nitric oxide precursors, L-arginine or beetroot extract as well as control, in aging humans that underwent a six-week training period. Overall, the manuscript has relatively minor editorial (grammar, language, etc.,) modifications that should be addressed. The authors should provide explicit hypothesis(es) tested as well as provide evidence for purity of supplements utilized. Even though these are among limitations in the manuscript’s present form, the authors should be able to address these points for full consideration of acceptance.
Answer: We thank to the Reviewer for his/her comments. Regarding English editing, weare not native English speakers. If the manuscript is accepted, we plan to contract the editorial services provided by MPDI to correct these mistakes. Regarding the supplements used, this point has been raised by the Editor (see lines 139-140). Following this section, we address all limitations raised by the Reviewer.
SPECIFIC:
TITLE
- None
ABSTRACT
- Page 1, Line 37: Change "...to the beginning." to "...to baseline."
Answer: Change has been performed accordingly (line 38).
- Page 1, Line 37: Add: ..."during short-term training while supplementing, preserving..." following "supplement"
Answer: Change has been performed accordingly (lines 38-39).
INTRODUCTION
- Pages 1-2, Line s 45-52: "Degeneration" has a very specific denotative meaning in medicine/physiology; and, while the author's note the disability and loss of skeletal muscle mass that occurs with the aging process, it is not putative that the "hallmarks" of the aging process include "degeneration". The word "deterioration" would be a better word choice.
Answer: Change has been performed accordingly (lines 46 and 52).
- Page 3, Line 108: The authors should state explicit/specific hypothesis(es)
Answer: This point has been raised by Reviewer-1. The specific statement for the hypothesis can be found at lines 87-99.
MATERIALS and METHODS
- Page 3, Line 129: Please explain/clarify this range for arterial hypertension given.
Answer: Healthy arterial tension is 120 mmHg. However, uncontrolled arterial hypertension has large fluctuations. We did not want to take any risk, this why we decided to put the lower limit at 100 mmHg.
- Page 3, Line 137: Include "to" between "prior intervention"
Answer: Typo has been corrected accordingly (line 128).
- Page 3, Line 143: Change "...for all of them." to "...for all participants."
Answer: Change has been performed accordingly (line 134).
- Page 3, Line 145: Change "From the beginning of the study and at the..." to "Initially, at the..."
Answer: Change has been performed accordingly (line 136).
- Page 4, Line 150: Were supplements/compounds USP grade? If not, how was the purity verified?
Answer: Yes, according to manufacturer information.
- Page 4, Line 175: Change "tries" to "attempts"
Answer: Change has been performed accordingly (line 167).
- Page 4, Line 192: Change "he" to "the researcher"
Answer: Change has been performed accordingly (line 184).
RESULTS
- Page 5, Lines 202-203: Change "...and the second, the last day." to "...the second day, and on the final day."
Answer: Change has been performed accordingly (lines 194-195).
- Page 5, Lines 203-205: The authors should describe how the blood draw site was prepared (i.e. sterile prep, etc.,) prior to extraction.
Answer: Blood extractions were performed in a laboratory certified by the Spanish Health System. Therefore, the site was following the Spanish legislation. This point has been indicated (line 196).
- Pages 5-6, Lines 229-230: What does "an almost healthy weight" signify?
Answer: A healthy BMI is 20-25 kg/m2. Participants have 25.2 kg/m2 (women) and 27.4 kg/m2 (men). The interpretation is that the main part of participants is into a healthy range for BMI, but some of them are not, but they are not undergoing obesity. This is why we mean when we say “an almost healthy weight, according to the BMI”.
- Page 7, Line 259: Delete "have"
Answer: Change has been performed accordingly (line 252).
- Page 7, Line 272: Change "rest of" to "remaining"
Answer: Change has been performed accordingly (line 265).
- Page 7, Line 273: Change "taking the supplements" to "supplemented"
Answer: Change has been performed accordingly (lines 265-266).
- Page 8, Line 293-294: Change "optimal" to "robust" or another term the authors feel is appropriate
Answer: Change has been performed accordingly (line 286).
- Page 8, Line 296: Change "All these positive scores..." to "Collectively, the positive scores..."
Answer: Change has been performed accordingly (line 289).
- Page 8, Line 305: Change "on" to "for"
Answer: Change has been performed accordingly (line 299).
DISCUSSION
- Page 9, Line 335: Change "...of the women and men who..." to "...in women or men who..."
Answer: Change has been performed accordingly (line 329).
- Page 9, Lines 340-341: How can the authors make this statement, for they did not measure intensity directly nor do they have a young group included in the current study. This must be restated.
Answer: We refer to adaptative and metabolic changes in general. We have changed the statement accordingly (lines 332-336).
- Page 10, Lines 345-349: Six weeks of exercise training may be too short of an exposure duration (possible limitation?).
Answer: This explanation has been included according to Reviewer suggestions (lines 345-346).
- Page 10, Line 368: Delete "all"
Answer: Change has been performed accordingly (line 360).
- Page 10, Line 368: Change "suggests" to "suggest"
Answer: Change has been performed accordingly (line 360).
- Page 10, Line 382: Change "...level of muscle use and damage" to "...level of muscle damage.". It does not indicate the "level of use", which would be indicative of activation.
Answer: Change has been performed accordingly (line 374).
- Page 11, Paragraph 1, Line 408: The authors should note/address the/any limitation(s) of the current study.
Answer: Limitations have been included (lines 401-405).
REFERENCES
- None
FIGURE LEGENDS/FIGURES
- None
Round 2
Reviewer 1 Report
Thanks for reply.
Reviewer 2 Report
The authors have addressed all of this reviewer's concerns and/or made all suggested revisions.